Resource

# Multi-region proteome analysis quantifies spatial heterogeneity of prostate tissue biomarkers

Tiannan Guo[1,2,*], Li Li[3,*], Qing Zhong[4,5,*], Niels J Rupp[4], Konstantina Charmpi[3], Christine E Wong[4], Ulrich Wagner[4], Jan H Rueschoff[4], Wolfram Jochum[6], Christian Daniel Fankhauser[7], Karim Saba[7], Cedric Poyet[7], Peter J Wild[4,8], Ruedi Aebersold[1,9], Andreas Beyer[3,10]

It remains unclear to what extent tumor heterogeneity impacts on protein biomarker discovery. Here, we quantified proteome intra-tissue heterogeneity (ITH) based on a multi-region analysis of prostate tissues using pressure cycling technology and Sequential Windowed Acquisition of all THeoretical fragment ion mass spectrometry. We quantified 6,873 proteins and analyzed the ITH of 3,700 proteins. The level of ITH varied depending on proteins and tissue types. Benign tissues exhibited more complex ITH patterns than malignant tissues. Spatial variability of 10 prostate biomarkers was validated by immunohistochemistry in an independent cohort (n = 83) using tissue microarrays. Prostate-specific antigen was preferentially variable in benign prostatic hyperplasia, whereas growth/differentiation factor 15 substantially varied in prostate adenocarcinomas. Furthermore, we found that DNA repair pathways exhibited a high degree of variability in tumorous tissues, which may contribute to the genetic heterogeneity of tumors. This study conceptually adds a new perspective to protein biomarker discovery: it suggests that recent technological progress should be exploited to quantify and account for spatial proteome variation to complement biomarker identification and utilization.

## Introduction

During the last decade, numerous new cancer treatment options have been developed. Their optimal application, however, requires better molecular characterization of the tumors with the aim of developing biomarkers matching the specific tumor to the best available therapy. Some cancer types, such as prostate cancer, still suffer from an "over-treatment problem," i.e., radical therapy such as removal of the organ in unnecessary cases because of uncertain diagnosis. These problems persist despite the recent progress in genomic, transcriptomic, and proteomic profiling of tumors. In contrast to the standardization of histopathological diagnostic categories, tumor grading, and standards of reporting, molecular testing is still underexploited in routine diagnostics of localized prostate cancer (PCa) cases. A recent review about biomarkers in prostate cancer (Kristiansen, 2018) has highlighted the need to consider intra-tissue heterogeneity (ITH) in each individual case for successful molecular testing. ITH is of high clinical relevance. For instance, a tumor may contain a small subpopulation of cells with primary resistance, leading to incomplete response to treatment or early recurrence (Murtaza et al, 2015). A high degree of heterogeneity in Gleason score, DNA ploidy, and phosphatase and tensin homolog expression has been observed in prostate tumors (Cyll et al, 2017). Thus, it remains a challenge to optimize clinical decisions based on single biopsies (Boutros et al, 2015).

Indeed, ITH is an important contributor to spatially variable molecular levels, which poses a substantial problem for biopsy-based tumor diagnostics because for highly variable proteins, the measured quantity is position dependent. Genomic ITH has been predicted based on clonal evolution and the cancer stem cell hypothesis (Dalerba et al, 2007). This prediction was experimentally validated by the application of high-throughput sequencing to small tissue samples and even single cells. Such studies have uncovered a high degree of genetic ITH in the colon (Jones et al, 2008), pancreas (Yachida et al, 2010), breast (Russnes et al, 2011), prostate (Haffner et al, 2013), renal carcinomas (Gerlinger et al, 2012), and leukemia (Ding et al, 2012; Cancer Genome Atlas Research N, 2013), with regard to both mutational and gene expression profiles of tumor cells. For example, Boutros et al (2015) observed extensive ITH in prostate cancers at the level of gene copy number

[1]Department of Biology, Institute of Molecular Systems Biology, Eidgenössische Technische Hochschule Zürich, Zurich, Switzerland [2]Institute of Basic Medical Sciences, Westlake Institute for Advanced Study, Westlake University, Hangzhou, Zhejiang, China [3]Cluster of Excellence Cellular Stress Responses in Aging-Associated Diseases, University of Cologne, Cologne, Germany [4]Departments of Pathology and Molecular Pathology, University Hospital Zurich, Zurich, Switzerland [5]Children's Medical Research Institute, Faculty of Medicine and Health, University of Sydney, Westmead, Australia [6]Institute of Pathology, Cantonal Hospital St. Gallen, St. Gallen, Switzerland [7]Department of Urology, University Hospital Zurich, University of Zurich, Zurich, Switzerland [8]Senckenberg Institute of Pathology, University Hospital Frankfurt, Frankfurt am Main, Germany [9]Faculty of Science, University of Zurich, Zurich, Switzerland [10]Center for Molecular Medicine Cologne, University of Cologne, Cologne, Germany

Correspondence: peter.wild@kgu.de; aebersold@imsb.biol.ethz.ch; andreas.beyer@uni-koeln.de
*Tiannan Guo, Li Li, and Qing Zhong contributed equally to this work.

alterations and point mutations, which led to spatially divergent mutational patterns for thousands of genes, including several tumor-relevant genes (Boutros et al, 2015). It can be expected that genomic ITH will be translated, at least to some extent, to ITH at the protein level. For example, androgen receptor and prostate-specific antigen (PSA)/ kallikrein 3(KLK3) expression can significantly vary between different regions within the same prostate carcinoma (Magi-Galluzzi et al, 1997; Shah et al, 2015). Thus, there is a need to systematically describe and quantify protein-level heterogeneity in tumor tissues.

Despite this well-recognized need, technical challenges have so far prevented the quantification of protein-level heterogeneity in tumor specimens at the proteomic scale (Alizadeh et al, 2015). High-throughput antibody-based immunohistochemistry (IHC) staining has been applied to tissue sections (Uhlen et al, 2015). However, such data are semiquantitative and limited in scope by the availability of suitable antibodies. Single-cell proteomics using mass cytometry is another promising technology allowing quantification of protein levels in thousands of individual cells. However, the technique at present only measures tens of proteins per sample (Giesen et al, 2014). Label-free shotgun proteomics has been used to compare the proteomes of three regions of colon tissues isolated by laser capture microdissection (Wisniewski et al, 2012). During the review of this study, Buczak et al (2018) reported quantitative proteomic comparison of five pairs of tumorous and non-tumorous microdissected formalin-fixed paraffin-embedded tissues from patients with hepatocellular carcinoma using 10-plex tandem mass tags (TMT; Thermo Fisher Scientific) and identified protein abundance changes between tumorous and peritumorous tissues, including NADH hydrogenase complex I, which is also observed as changed in 11 murine hepatocellular carcinoma tumors compared with normal murine livers using label-free quantification (Buczak et al, 2018). In another experiment of three concentric sector regions, a tumor capsule region, a peritumoral tissue region, and the bulk tumor, the authors quantified 2,698 Uniprot proteins (excluding protein groups) using 6-plex tandem mass tags and 2,166 proteins using data-independent acquisition. This study found that most of the quantified proteins were expressed at comparable levels across the whole specimen and detected abundance changes of multiple proteins across regions including collagens, fibrillin, and decorin. The authors also identified consistency between proteome and transcriptome data in terms of gene expression changes, implying that spatial heterogeneity is largely driven by protein synthesis variation.

Despite this progress, it remains important to separate technical variability from true spatial ITH and to investigate the relationship between inter-individual heterogeneity and intra-tissue heterogeneity. Answering these questions requires a rigorously designed study, a highly reproducible proteomics technology, the ability to analyze multiple regions of a bulk tumor, and statistical models to deconvolute various types of protein variation.

We have recently developed a mass spectrometry–based proteomics method, i.e., pressure cycling technology (PCT) and Sequential Windowed Acquisition of all THeoretical fragment ion mass spectra (SWATH) (Guo et al, 2015a), which supports highly reproducible and accurate quantification of a few thousand proteins from biopsy-scale tissue samples at high throughput. This is accomplished by integration into a single platform of optimized sample preparation, and mass spectrometric and computational elements. To generate mass spectrometry–ready peptide samples from tissue samples, we adopted PCT to lyse the tissues, extract proteins, and digest them into peptides in a single tube under precisely controlled conditions (Powell et al, 2012). To analyze the resulting peptide samples, we used SWATH-MS, a massively parallel targeting mass spectrometry method (Gillet et al, 2012). In SWATH mass spectrometry (SWATH-MS), all MS-measurable peptides in a sample are fragmented and periodically recorded over a single dimension of relatively short chromatography (Gillet et al, 2012). The net result of this technique is a single digital file that contains fragment ions of all mass spectrometry–detectable peptides, from which peptides and proteins are identified and quantified post-acquisition, via a targeted data analysis strategy (Gillet et al, 2012; Röst et al, 2014).

In this study, we approached proteomic ITH for prostate cancer tissues by PCT-SWATH–based multi-region proteomic analysis of 60 biopsy-level tissue samples from three prostate cancer patients. We then computed the technical and spatial biological variation for each measured protein in different types of tissues and different patients, and established a proteome-scale landscape of protein ITH in benign and malignant prostate tissues. Our data revealed distinct ITH patterns of prostate cancer biomarkers that were further independently validated using IHC in an independent set of 83 patients.

## Results

### Study design for quantifying proteomic variability

We designed a study to quantify spatial proteomic variability in multiple regions of malignant and matching benign prostate tissues using the PCT-SWATH-MS platform (Guo et al, 2015). We assumed that the total proteomic variability observed in the sample cohort was composed of technical and biological variation, the latter including inter-patient, inter-tissue, and intra-tissue variation. To open the possibility to partition the overall observed variability into its possible sources, we obtained tissue samples from multiple regions of prostatectomy specimens as illustrated in Fig 1A. Each sample was a tissue punch biopsy consisting of a cylinder of 1-mm diameter and about 3-mm length that was derived from fresh frozen tissue blocks using a core needle. The samples were obtained from prostatectomy specimens in three individuals diagnosed with adenocarcinoma (ADCA) of the prostate. Gleason grading was performed according to the International Society of Urological Pathology and the World Health Organization consensus (Epstein et al, 2016; Humphrey et al, 2016) (Fig S1). In total, 12 benign prostatic hyperplasia (BPH) and 18 ADAC tissue samples were obtained. One of the three individuals had a mixed acinar and ductal ADAC, and both subtypes were included in the study to measure the variation resulting from morphologically distinct subtypes. The other two patient samples displayed acinar ADCA by histologic means. Each tissue type (malignant versus benign) of each patient was sampled three to six times, resulting in a total of 30 biological samples. Each sample was processed by

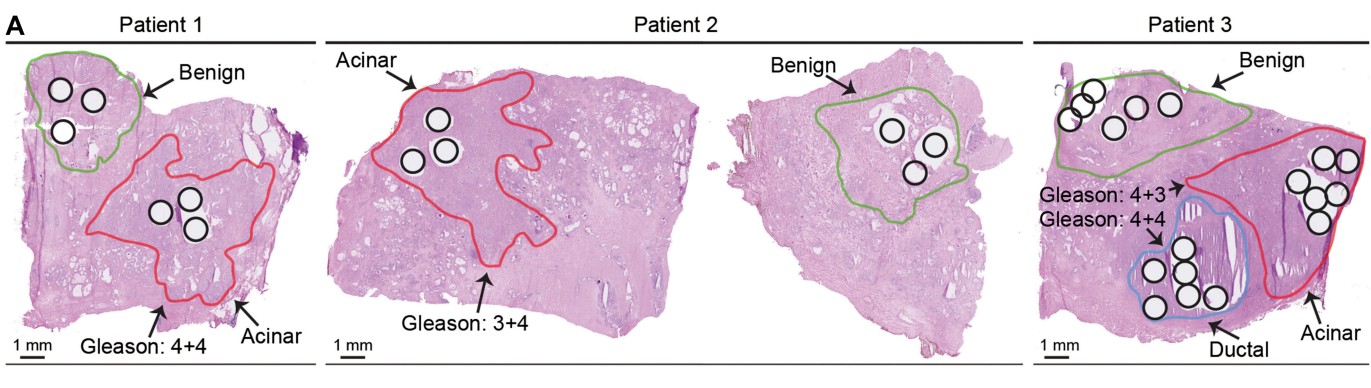

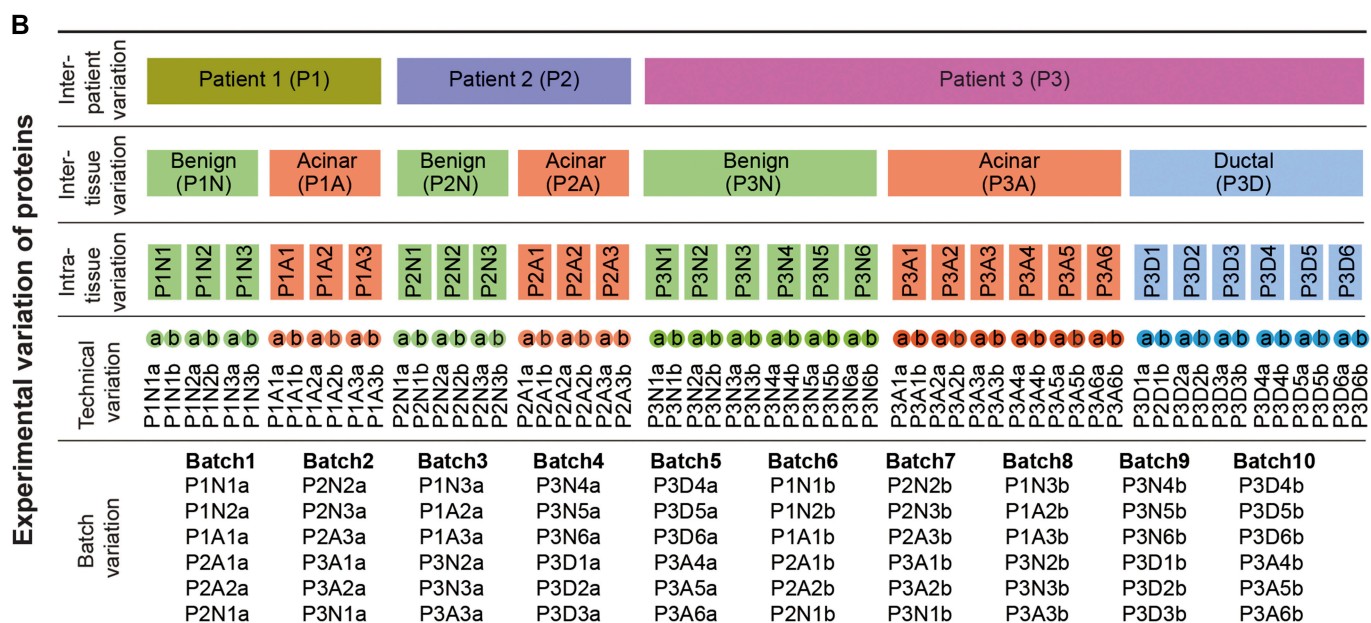

**Figure 1. Study design.**
**(A)** Hematoxylin and eosin staining of the fresh frozen prostate tissue from three individuals who have contributed to BPH (non-tumorous) and matching acinar or ductal ADCA. Green, orange, and blue lines depict regions diagnosed by a pathologist as BPH, acinar, and ductal tumors, respectively. Black circles indicate where the punches were made. **(B)** Overall measured variation of protein expression was partitioned into biological and technical variation including inter-patient variation, inter-tissue variation, intra-tissue variation, and technical variation from MS analysis and batch variation. Three or six punches were sampled from each tissue type, followed by PCT-SWATH analyses in technical duplicate. The samples were shuffled and analyzed in 10 batches of six samples.

PCT-SWATH in duplicate to evaluate the technical variation of the proteomic analysis (Fig 1 and Table S1). The samples were grouped into 10 batches of six samples, according to patient identity, tissue type, and technical replicate (Fig 1B and Table S2). This experimental design allowed us to subsequently estimate intra-tissue variability from within-batch comparisons (see the Materials and Methods section), which is important to avoid overestimating variances due to batch effects.

### Quantitative proteomic analysis of 30 prostate cancer tissue regions

The 10 batches of samples were processed using PCT-SWATH in duplicate over a period of 15 working days. The acquired SWATH-MS data were subjected to in silico–targeted analysis using the OpenSWATH software (Röst et al, 2014). In total, 36,660 proteotypic peptides and 6,873 proteins were quantified consistently across all 60 measurements (Tables S3 and S4). The measured protein intensities were highly reproducible (average Pearson correlation values between replicates: 0.944). To obtain high-confidence estimates of ITH, we subsequently narrowed our statistical variation analyses to a subset of 3,700 proteins quantified by at least two concordant proteotypic peptides. Our peptide selection procedure ensured that the selected peptides showed consistent behavior across samples. Thereby, we minimized the possibility that peptide intensity variation was not due to protein abundance changes but due to posttranslational modifications (PTMs) or other artifacts (see the Materials and Methods section) (Picotti et al, 2013). We then

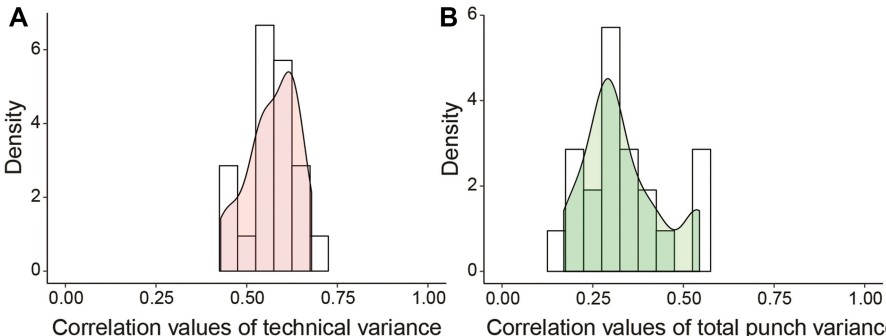

**Figure 2. Consistency of technical and total variance.**
**(A)** Correlation of technical variances estimated independently for different samples. Technical variance is estimated from technical replicates.
**(B)** Correlation of total variances (between punches) estimated independently from punches from different tissue samples (different patients, different tissue types).

corrected batch effects in the dataset by subtracting the average signal of each protein per batch. After batch correction, most technical replicates grouped together by unsupervised clustering based on the abundance of all proteins (Fig S2).

### Quantification of spatial proteomic heterogeneity

Our estimates of proteomic ITH are based on the notion that the signal variation between two samples is due to a combination of biological and technical factors. Because the biological variation is not directly quantifiable, we estimated biological variance by subtracting the technical variance from the total observed punch-to-punch variance.

The technical variance was estimated by calculating the dispersion between two technical replicates for each sample (independent protein digests from the same punch measured separately), i.e., generating 30 initial technical variance estimates per protein before averaging them (see the Materials and Methods section for details). This strategy produced seven technical variance estimates for all pairs of patient/tissue type (three normal tissue regions, three acinar tissue regions, and one ductal tissue region; Fig 1). Pairwise correlations of these seven independent estimates showed that technical variances were consistently positively correlated, with a median correlation of 0.572 (Fig 2A). Likewise, we analyzed the same type of correlation for the total punch variances. Like the technical variance, independent estimates of the total variance were also highly correlated, albeit with a slightly lower median correlation of 0.302, suggesting that the technical variance was more robust and less dependent on the specific sample than the total variance and the biological variance (Fig 2B). Thus, as expected, the technical variance of a protein was mostly determined by its physicochemical properties, whereas total variance varied in different tissue samples, probably because of biological factors. Furthermore, technical variance of log-transformed intensities was independent of the mean log-intensity (Fig S3), suggesting that the same estimate of technical variance could be used at high and low protein concentrations. Subsequently, we averaged the seven estimates of technical variance per protein to obtain a single, robust estimate of each protein's technical variance.

Having established that our estimates of total variances and technical variances are robust, we next computed biological variances by subtracting each protein's technical variance from its total variance between punches of the same patient and tissue type (see the Materials and Methods section). This yielded an estimate of intra-tissue biological variances of protein abundance that can be interpreted as the degree of proteomic ITH. The technical and total variances were independently estimated, which makes it numerically possible that the technical variance can be larger than the total variance of a specific set of punches. Indeed, for 183 proteins (4.9%), the estimated technical variance was larger than the total variance (Fig S4). These were mostly the proteins with very low total variance. We could not rigorously quantify the biological variances of these proteins; nevertheless, we assumed that most of them would have comparably low biological variances. Proteins with technical variances higher than total variances were excluded from most subsequent analyses.

Next, we compared the biological variances within a tissue with the biological variance between tissue types (benign versus malignant; termed "inter-tissue") and between patients (Fig 3). Inter-tissue and inter-patient variances were obtained by first averaging protein intensities from punches of the same tissue or patient, respectively (Fig 1A and see the Materials and Methods section). Our data showed that the biological variance between punches within the same tissue (i.e., intra-tissue variance) is of similar magnitude as the variation of average intensities between tissues and patients, indicating a high degree of protein ITH (Fig 3A). Furthermore, the protein variances between patients, between tissue, and within tissue were significantly correlated (Fig 3B–D). Thus, a protein with large intra-tissue variation is also likely to vary across tissues and between the three patients.

### Classification of proteins based on their intra-tissue variability

To characterize ITH in different tissue types, we compared the biological variance of each protein in benign and malignant prostate tissues, and quantified the variability of 3,517 proteins in BPH and ADCA tissue samples (Table S5). Interestingly, we observed a strong dependence of the variability of some proteins on the tissue type. We then classified the thus quantified proteins into five groups based on their biological variance patterns in the different sample types (Fig 4A). Group 1 consisted of 100 proteins that were always robust and generally showed little intra-tissue variation in benign and malignant prostate tissues. Group 2 consisted of 339 proteins that varied substantially more in benign tissues compared with malignant tissues. Group 3 consisted of 93 proteins that varied

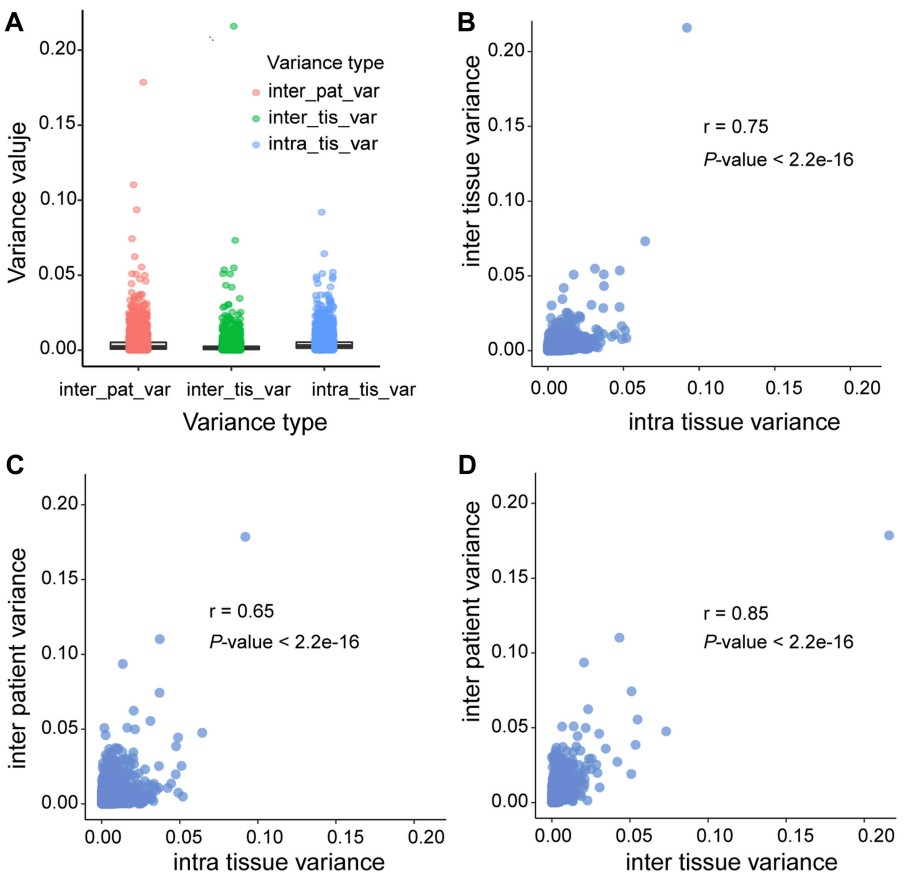

**Figure 3. Correlation of biological variance between patients and tissue types.**
Each dot represents one protein. **(A)** Distributions of biological variance estimates. Inter-patient variances and inter-tissue variances are based on averaging the measurements of at least three punches. Intra-tissue variance was first determined independently per patient and tissue type, and then averaged. **(B)** Biological variance between tissue of the same patient versus variance between punches of the same patient and tissue. **(C)** Biological variance between different patients but same tissue type versus variance between punches of the same patient and tissue. **(D)** Biological variance between the same tissue types in different patients versus variance between different tissue types of the same patient.

more strongly in malignant tissues compared with benign tissues. Group 4 contained 365 proteins that had high intra-tissue variance in both malignant and benign tissues, whereas Group 5 contained the remaining 2,620 proteins with intermediate variability. Remarkably, the top three most variable proteins in BPH are the three proteins known or used in the diagnosis of prostate tumors, including PSA/KLK3, prostatic acid phosphatase (PAP)/acid phosphatase, prostate (ACPP), and Desmin (DES). PSA is an androgen-regulated kallikrein family serine protease that is produced by the secretary epithelial cells in acini and ducts of prostate glands (Balk et al, 2003). The secreted PSA, originated from prostate tissues, is the most commonly used blood-based biomarker for prostate cancer (Hayes & Barry, 2014). However, PSA screening has remained controversial because of uncertainty surrounding its benefits and risks and the optimal screening strategy (Barry, 2009). Our data showed that PSA in situ was most variable in BPH but more stable in ADCA tissues. Because PSA is regulated by androgen, this indicates androgen-driven malignant growth of prostate tumor cells. PAP is a nonspecific tyrosine phosphatase and a well-studied tumor suppressor for PCa. PAP has already been used in immunotherapy regimens against PCa (Di Lorenzo et al, 2011) and is the second most variable protein in BPH after PSA. The variability of PAP expression was relatively high in ADCA samples but lower than its variability in BPH samples. DES constructs class-III intermediate filaments in smooth muscle cells. As a marker for prostate stromal composition, DES expression has already been associated with PCa

survival (Ayala et al, 2003). Tuxhorn et al (2002) have shown that prostate cancer–reactive stroma is composed of a myofibroblast/fibroblast mix with a significant decrease or complete loss of fully differentiated smooth muscle, whereas normal prostate stroma is predominantly smooth muscle (Tuxhorn et al, 2002). Given the known heterogeneous composition of myoglandular hyperplasia (i.e., BPH) out of glandular and stromal (smooth muscle) elements, the higher variability of DES expression in BPH compared with PCa is not surprising.

To further investigate the protein variability classes, we then performed a gene ontology (GO) enrichment analysis (Fig 4B). As expected, stable proteins of Group 1 were enriched for basic cellular functions that were required irrespective of the tissue state, such as energy metabolism (Fig 4B). Proteins highly variable in both malignant and benign tissues (Group 4) were enriched for immunity-associated processes. Muscle-related proteins exhibited a high degree of heterogeneity in benign tissues, reflecting the fact that smooth muscle fibers are part of healthy prostate tissues, whereas prostate cancer glands are per definition closely packed with less intervening stroma (Humphrey et al, 2016). This agrees with the variability observed for the DES as discussed above. Proteins associated with cell cycle–related functions such as nucleosome and chromatin assembly displayed a high degree of heterogeneity in malignant tissues. Thus, our data are consistent with recent findings, suggesting that the proliferation rates among prostate cancer cells can be highly variable (Zellweger et al, 2009) and that

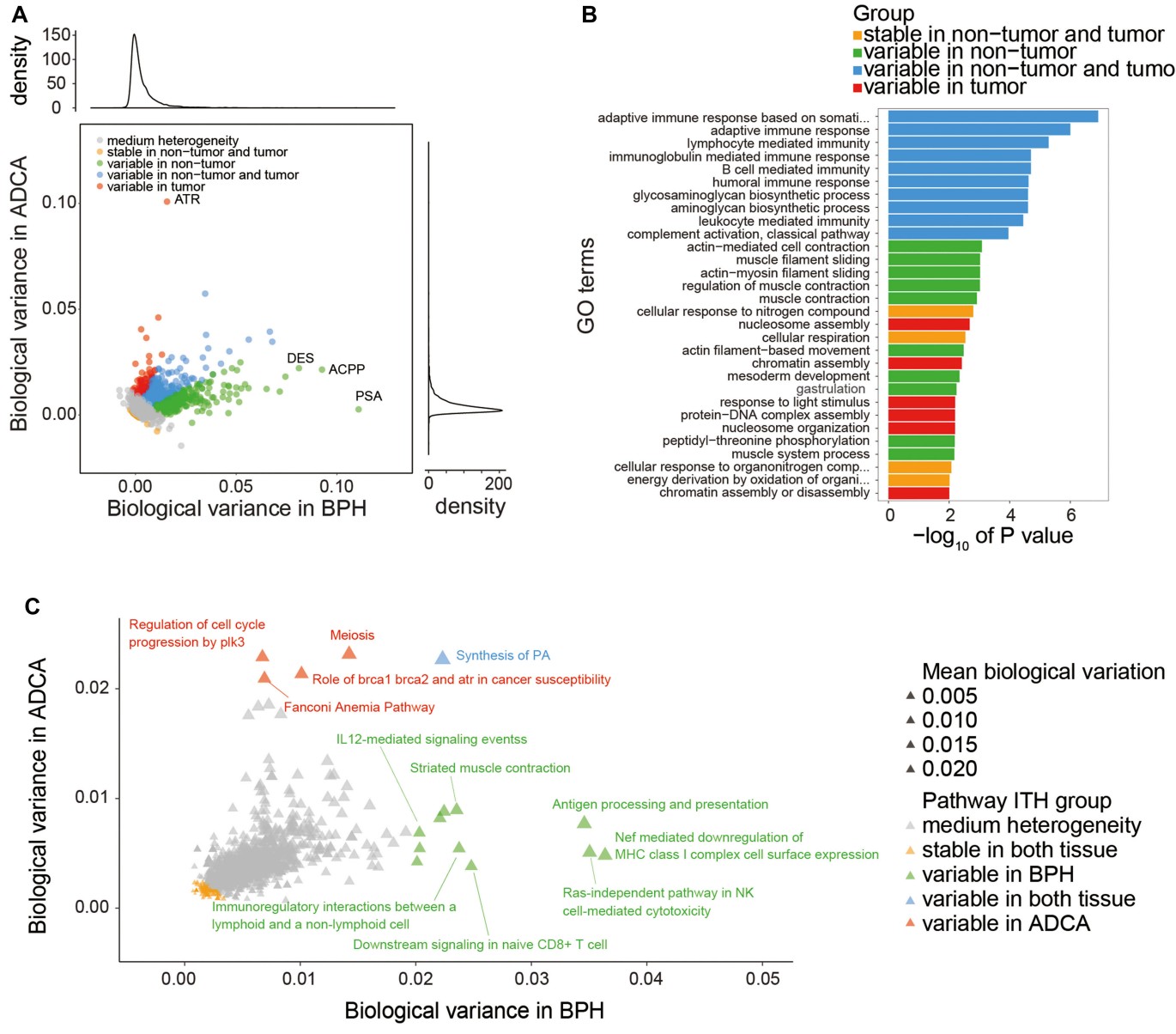

**Figure 4. Intra-tissue heterogeneity in tumorous and non-tumorous tissue.**
**(A)** Biological variance among punches from the same tissue region was considered as the degree of intra-tissue heterogeneity for the respective tissue type. Degree of intra-tissue heterogeneity for each protein in benign versus malignant tissue are shown and colored according to classification. **(B)** GO enrichment analysis of four protein categories from (A). Length of horizontal bars indicates the significance of the enrichment. **(C)** Intra-tissue heterogeneity of biochemical pathways. Each triangle is the average biological variance (intra-tissue heterogeneity) of all quantified proteins from the respective pathway. Degree of intra-tissue heterogeneity for each pathway in benign versus malignant tissue is shown. Pathways were grouped according to their variability in benign and malignant tissue.

epigenetic events are of high importance in prostate carcinogenesis (Grasso et al, 2012; Plass et al, 2013; Beharier et al, 2015).

**Spatial heterogeneity of biochemical pathways**

Based on the determined protein-level variance patterns described above, we could also interrogate the ITH of biochemical pathways. To quantify a pathway's variance, we computed the average biological variance (intra-tissue variance) for all human pathways from ConsensusPathDB (Kamburov et al, 2013) with at least five

quantified proteins (Fig 4C). Like the individual proteins, we grouped pathways into five groups depending on their degrees of heterogeneity in malignant and benign tissues. Five pathways emerged as being particularly variable in tumor tissues (i.e., average biological variance in malignant samples above 0.02): "Fanconi Anemia Pathway," "Meiosis," "Meiotic synapsis," "Regulation of cell cycle progression by plk3," and "Role of brca1, brca2, and atr in cancer susceptibility." These pathways are involved in DNA damage response and include proteins such as serine/threonine protein kinase ataxia telangiectasia and Rad3-related protein (ATR)

and the cohesion complex. The specific role of these pathways in responding to chromosomal aberrations suggests that the occurrence and repair of double strand breaks (which are a hallmark of prostate cancer) are heterogeneous within tissue specimens (Haffner et al, 2010). Pathways highly variable only in non-tumorous tissues are markedly enriched for immune activity. The stromal component of BPH samples demonstrated a high degree of ITH in antigen processing and presentation, naive CD8+ T cell signaling, IL-12–mediated signaling, interactions between a lymphoid and a nonlymphoid cell, MHC class I complex expression, and natural killer-cell–mediated cytotoxicity, suggesting the combat between carcinogenesis and immunity. Consistent with the previous analysis, we observed more variable muscle contraction activity in non-tumorous tissues. The only pathway variable in both tumorous and non-tumorous tissues was the synthesis of phosphatidic acid, a critical component of mammalian target of rapamycin signaling and a biosynthetic precursor for all cellular acylglycerol lipids with critical roles in prostate tissue biology (Fang et al, 2001; Foster, 2009).

### Investigation of spatial heterogeneity of selected proteins using IHC in an independent cohort

We further investigated the biological variation of selected proteins from the PCT-SWATH analysis using a complementary technology in an independent, larger cohort. We constructed a tissue microarray (TMA) using benign and malignant (ADCA) prostate tissues from 83 additional patients and established IHC assays to measure the expression of 10 representative proteins in the various ITH groups identified from the PCT-SWATH results, including actin related protein 1 homolog B (ACTR1B), DES, PSA, and growth/differentiation factor 15 (GDF15) as shown in Fig 5, as well as ACPP, ABCF1, NUP93, CUTA, CRAT, and FSTL1 (Fig S5). This set of validation proteins contains some well-established markers for prostate cancer to elucidate their variability within benign and tumorous tissue specimens. The stained TMAs contained duplicate tissue cores of 48 ADCA and 35 BPH samples. The heterogeneity of proteins was evaluated based on an immunoreactivity score computed from duplicate tissue spots and measured by using the Pearson correlation coefficient between the two spots for BPH and ADCA (Fig 5). Thus, a high Pearson correlation score indicates a homogeneous distribution of the respective protein in the TMAs (i.e., low ITH). We found that the degree of ITH determined in the three patients by PCT-SWATH was well validated in the independent cohort. ACTR1B is an actin-related protein in the dynamin complex to construct cytoskeleton. This housekeeping protein exhibited a very high degree of correlation in both BPH ($r = 0.96$) and ADCA ($r = 0.80$) samples, serving as a positive control. In the TMA cohort, DES was more variable in BPH ($r = 0.51$) than in ADCA ($r = 0.67$), which is consistent with proteomics data. Our TMA data demonstrated that in BPH samples, PSA was found only in the glandular tissue and expressed more heterogeneous than in ADCA samples, with blood PSA levels being a nonspecific biomarker for PCa. GDF15 is a stress-induced cytokine belonging to the transforming growth factor beta superfamily (Vanhara et al, 2012). This protein is expressed in highly complex forms with distinct biological functions related to immunity. In various tumors, including prostate cancer, GDF15 interacts with the extracellular matrix and promotes tumor progression and metastasis (Vanhara et al, 2012). We found GDF15 to be expressed at relatively low levels in BPH with a low degree of ITH probably because of inflammatory changes of glandular architecture followed by stromal tissue increase in BPH (Vanhara et al, 2012). In the ADCA samples, GDF15 expression was elevated with a high degree of variation, indicating complex interactions between tumor cells and the microenvironment via modulators including GDF15. The high variability of ACPP in BPH samples was also confirmed in this cohort. Proteins grouped as medium heterogeneity, including ABCF1, NUP93, CUTA, CART, and FSTL1, displayed consistent heterogeneity patterns after manual inspection of the TMA data. Taken together, we observed significant correlations between the heterogeneity measured in the TMAs and the biological variance measures obtained with PCT-SWATH across all 10 proteins (Fig 6A and B).

## Discussion

This study investigated the spatial variability of the prostate proteome, which serves as a basis for better understanding the biology of PCa protein biomarkers. Protein biomarkers including PSA and GDF15 have been well studied in PCa; however, their spatial expression in prostate tissues has not been systematically studied. ITH has been studied at the morphologic and genomic level in diverse cancers, and it poses a major challenge for cancer biology and diagnosis (Alizadeh et al, 2015). However, proteomic ITH remains underexplored in prostate cancer, despite the critical roles of proteins in tumorigenesis and cellular biochemistry in general and the various single cell–based methods.

This study represents a technical advance toward understanding spatial ITH at the proteome level for solid tumors and other tissues. Using the PCT-SWATH methodology (Guo et al, 2015) and an associated data analysis strategy (Röst et al, 2014), we achieved deep proteomic coverage (consistent quantification of 6,873 reviewed SwissProt proteins across the 60 prostate tissue samples) and performed quantitative analysis of spatial ITH of 3,700 proteins, which were quantified by at least two proteotypic peptides that showed consistent abundance across samples. Despite the rigorous filtering, we could quantify a three times higher number of proteins than a recent proteomic analysis of primary prostate tissue samples (Iglesias-Gato et al, 2016). The number of proteins quantified in our study exceeds by one to two orders of magnitude the number of proteins typically quantified by tissue staining, which is the current standard method for protein quantification in clinical tissue samples. Our workflow is also compatible with laser capture microdissected samples, which can also be analyzed by using shotgun proteomics (Grosserueschkamp et al, 2017; Buczak et al, 2018; Garcia-Berrocoso et al, 2018). Our data did not achieve single-cell resolution like the mass cytometry (CyTOF) technology. These technologies, however, quantify orders of magnitude fewer proteins (Amir el et al, 2013; Giesen et al, 2014; Levine et al, 2015). The data generated in this study are unique with respect to the structure of the sample set, the degree of proteomic coverage, and the degree of measurement reproducibility and accuracy. Nevertheless, new MS-based proteomics technologies enabling analysis of single cells from tissue samples will be desirable to quantify

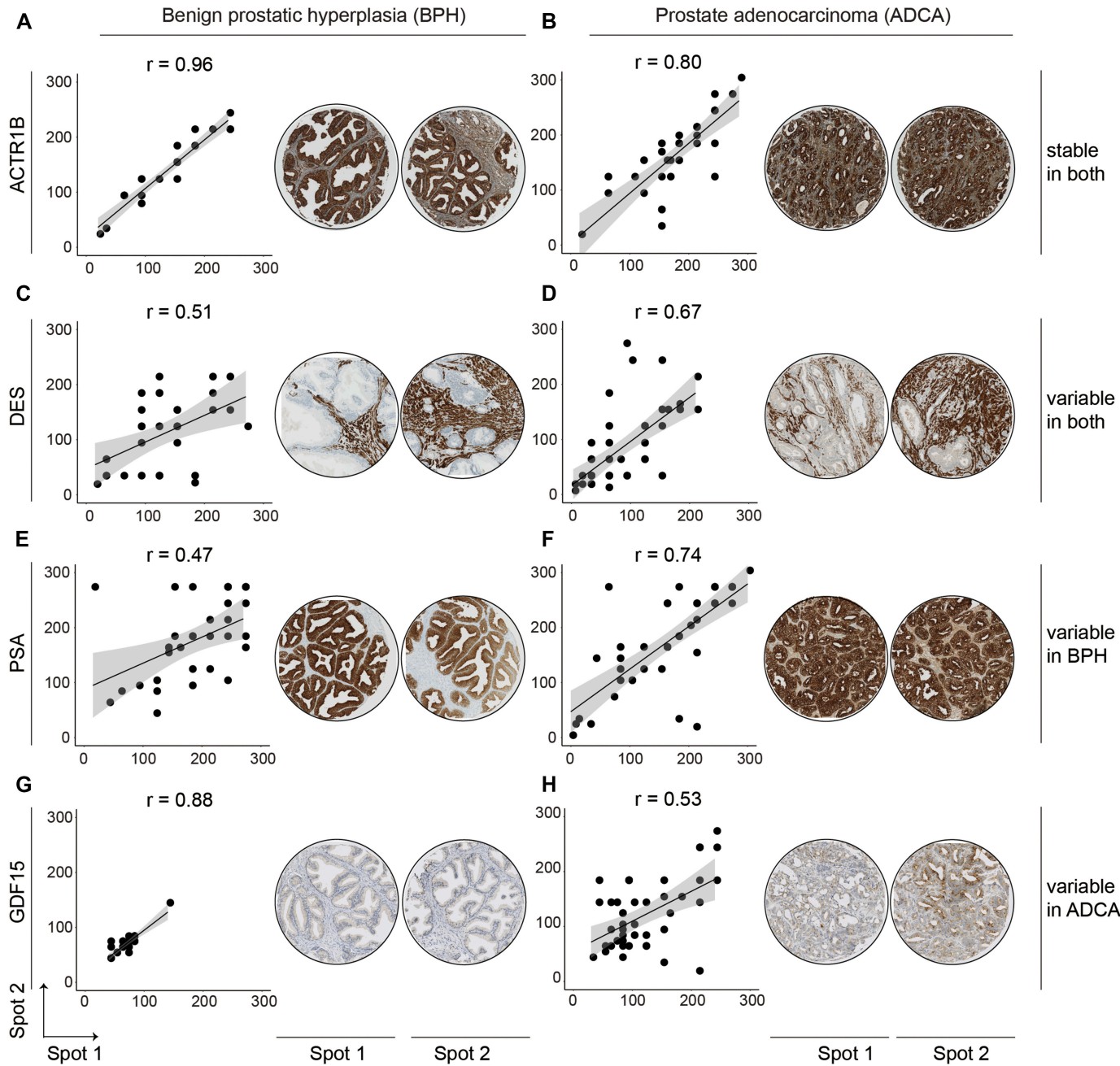

**Figure 5. Immunohistochemical validation of representative proteins.**
The top proteins from four ITH groups in BPH and malignant (ADCA) prostate tissue were validated using a TMA with two representative tissue spots of each patient.

spatial ITH at higher spatial resolution in future studies. Advanced matrix-assisted laser desorption/ionization imaging emerges as a useful tool to dissect ITH of proteins, peptides, and small molecules with high spatial resolution (Balluff et al, 2015; Widlak et al, 2016); however, the proteome depth and precision remains to be further improved.

The main goal of this study was not to discover new protein biomarkers; instead, we aimed to characterize the spatial ITH of the prostate proteome and investigate whether the ITH influences the utility of protein biomarkers and candidates. Our data contributed

to the understanding of the following prostate cancer biology. First, we systematically reported the degree of ITH of 3,700 SwissProt proteins in prostate tissues. Although some of these proteins are widely used in clinic, their expression pattern in prostate tumors was unclear. We found that PSA was preferentially variable in BPH, whereas GDF15 tended to vary in different tumor regions. This finding, together with the ITH pattern of eight more clinically relevant protein biomarkers, was further investigated and confirmed in an independent cohort of 83 PCa patients using TMA technology. This additional cohort analysis not only confirms that the

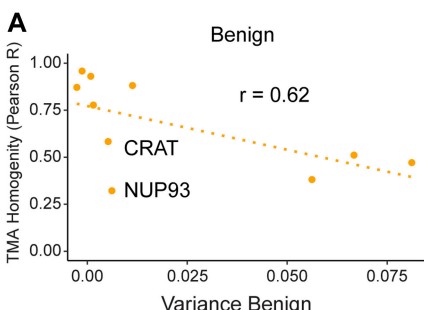

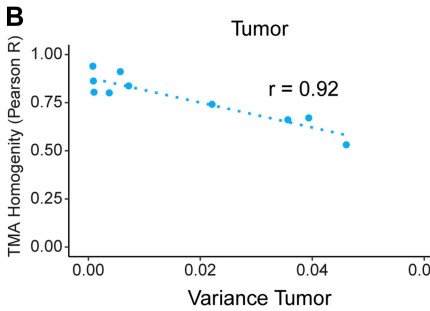

**Figure 6. Correlation between mass spectrometry–based (MS) variance estimates and TMA homogeneity.**
(A) shows benign tissues whereas (B) depicts tumor tissues. The concentrations of CRAT and NUP93 were almost zero in the benign tissue samples. Thus, it is virtually impossible to estimate their intra-tissue variation in benign tissues. The correlation between MS-based variance and TMA homogeneity was, however, computed without excluding these two proteins. NUP93 was slightly off the regression curve because its signal in IHC was relatively weak.

PCT-SWATH technology is a valid and practical extension of IHC and TMA for proteome-scale ITH analysis of clinical tissue samples but also consolidated the spatial variability of these proteins in prostate tissues, providing guidance for clinical application of these proteins as biomarkers. We found that protein ITH patterns vary between tissue types because of their biological functions and interplay with the microenvironment. Despite the high consistency with respect to ITH measured with TMA and proteomics, the two assays are of course not identical. For example, secreted proteins will likely be lost during IHC procedures, which partly explains the small discrepancy between TMA and MS data. Furthermore, protein truncations and other PTMs may have different effects depending on whether antibody binding and/or peptides quantified in the MS are affected.

Second, the data also shed light on the heterogeneity of multiple biochemical pathways. Interestingly, benign tissue displayed a high degree of variability in immunity-related signaling pathways, whereas tumor tissues, characterized by enhanced proliferation and DNA damage, exhibited high degree of heterogeneity in several DNA damage response pathways, suggesting that spatially variable DNA repair pathways probably contributed to genomic heterogeneity during the evolution of prostate cancers. We quantified the degree of ITH of several key proteins involved in DNA damage response, including ATR, MRE11, RAD21, RAD23A, RAD23B, RAD50, RAD9A, CHEK1, XRCC5, and XRCC6. The data showed that ATR, a DNA damage sensor, is variable only in tumors. We identified more proteins in the 6,873 proteins (3,700 proteins with two or more proteotypic peptides, and the other proteins with at least one proteotypic peptide), including BRCA2, ATM, RAD51C, RAD51AP2, XRCC1, XRCC2, and XRCC4. These proteins were not included in the ITH analysis because either single-proteotypic peptide identification or discordant quantity of multiple proteotypic peptides in a protein failed to pass our stringent inclusion criteria. Furthermore, we found that the degree of intra-tissue variability of multiple pathways was slightly higher in benign specimens compared with malignant tissues (Fig 4), which may be due to the more complex structure of healthy tissues involving a larger number of distinct cell types, whereas in tumorous tissues, most cell types are replaced by tumor cells.

The observed intra-tissue protein variability patterns have implications that extend beyond the present study to protein biomarker studies in general and have specific significance for biomarker studies in the context of personalized medicine, where sample availability is generally sparse. Our data suggest that the variation of some protein levels between patients is similar in magnitude to the variation within a single prostate. These findings underline the significance of low intra-tissue variability as an important property of a clinical protein biomarker. In fact, the observed variability patterns provide a rational explanation why some previously published tissue biomarker studies did not produce concordant results. Similar conclusions were drawn in an earlier study, in which the abundance variability of plasma proteins was analyzed in a twin cohort (Liu et al, 2015). The data indicated that those biomarker candidates that were proposed in the literature and eventually approved for clinical use showed low levels of variability derived from genetic differences in a population. In contrast, biomarker candidates proposed in the literature that showed a high degree of genetically caused abundance variation in a population were rarely validated. Our data add a new perspective to this problem: a candidate biomarker may show high variability between patients when quantified using single-needle biopsies per patient. However, the tumor-wide average concentrations may not be substantially different, and the true cause of the apparent inter-patient variability may be ITH, rather than rooted in the biochemical difference between normal and tumor tissues. Therefore, we suggest that intra-tissue variability of a protein or a pathway be used as an important criterion for the assessment of protein biomarker candidates, in addition to other parameters such as expression level and biochemical function. Including more biological replicates per patient to average out protein ITH or increasing patient numbers to account for variability may not always be possible. Thus, our work provides an important lead as to how ITH can be tackled even for small patient and sample numbers in clinically realistic scenarios.

# Materials and Methods

### Patients and samples for PCT-SWATH analyses

The prostates from three patients after prostatectomy were cut into tissue sections (thickness: about 3 mm). Fresh BPH and ADCA tissue sections were frozen and embedded in Tissue-Tek optimal cutting temperature compound (Sakura). The tissues were examined by trained pathologists and graded similarly according to the Gleason system as shown in Fig 1. Tumorous tissues from each patient contained acinar prostate tumors, whereas one patient

included an extra ductal prostate tumor. To obtain biopsy-scale tissue samples for PCT-SWATH analysis, we used a needle to punch out tissue cylinders (diameter: 1 mm, length: ~3 mm, wet weight: ~2 mg) at the locations as shown in Fig 1. About 100 µg proteins and 50 µg peptides were extracted per milligram of tissue. Multiple (three or six) punches were obtained from each area. The Ethics Committee of the Canton of Zurich approved all procedures involving human fresh frozen material. All three patients were part of the Zurich prostate cancer outcomes cohort study (ProCOC, KEK-ZH-No. 2008-0040) (Umbehr et al, 2008; Wettstein et al, 2017), and each patient signed an informed consent form.

## PCT-SWATH

The tissue samples were first washed to eliminate optimal cutting temperature, followed by PCT-assisted tissue lysis and protein digestion, and SWATH-MS analysis, as described previously (Guo et al, 2015). Briefly, each tissue punch was washed with 70% ethanol/30% water (30 s), water (30 s), 70% ethanol/30% water (5 min, twice), 85% ethanol/15% water (5 min, twice), and 100% ethanol (5 min, twice). Subsequently, the tissue punches were placed in PCT-MicroTubes with PCT-MicroPestle and 30 µl lysis buffer containing 8 M urea, 0.1 M ammonium bicarbonate, complete protease inhibitor cocktail (Roche) using a barocycler (model NEP2320-45k; Pressure BioSciences), which offers cycling alternation of high pressure (45,000 psi, 50 s per cycle) and ambient pressure (14.7 psi, 10 s per cycle) for 1 h. The extracted proteins were then reduced and alkylated before lys-C and trypsin-mediated proteolysis under pressure cycling. Lys-C (Wako; enzyme-to-substrate ratio, 1:40)–mediated proteolysis was performed under 45 cycles of pressure alternation (20,000 psi for 50 s per cycle and 14.7 psi for 10 s per cycle), followed by trypsin (Promega; enzyme-to-substrate ratio, 1:20)-mediated proteolysis using the same cycling scheme for 90 cycles. The resultant peptides were cleaned by Sep–Pak C18 (Waters Corp) and analyzed, after spike in 10% iRT (retention time) peptides, using SWATH-MS following the 32-fixed-size-window scheme as described previously with a 5600 TripleTOF mass spectrometer (SCIEX) and a 1D+ Nano liquid chromatography system (Eksigent). The liquid chromatography gradient was formulated with buffer A (2% acetonitrile and 0.1% formic acid in HPLC water) and buffer B (2% water and 0.1% formic acid in acetonitrile) through an analytical column (75 µm × 20 cm) and a fused silica PicoTip emitter (New Objective) with 3-µm 200-Å Magic C18 AQ resin (Michrom BioResources). Peptide samples were separated with a linear gradient of 2% to 35% buffer B over 120 min at a flow rate of 0.3 µl·min$^{-1}$. Ion accumulation time for MS1 and MS2 was set at 100 ms, leading to a total cycle time of 3.3 s.

## SWATH assays for prostate tissue proteome

We also analyzed unfractionated prostate tissue digests prepared by the PCT method using data-dependent acquisition (DDA) mode in a tripleTOF mass spectrometer over a gradient of 2 h as described previously (Röst et al, 2014). We spiked iRT peptides (Escher et al, 2012) into each sample to enable retention time calibration among different samples. We then combined this library with the DDA files from pan-human library (Rosenberger et al, 2014). Altogether, we analyzed 422 DDA files using X!Tandem (MacLean et al, 2006) and

Open Mass Spectrometry Search Algorithm (Geer et al, 2004) against a target-decoy, nonredundant human UniProtKB/SwissProt protein database (October 21, 2016) containing 20,160 protein sequences and the iRT peptide sequences. Reversed protein sequences were used as decoy sequences. We allowed maximal two missed cleavages for fully tryptic peptides, 50 ppm for peptide precursor mass error, and 0.1 Da for peptide fragment mass error. Static modification included carbamidomethyl at cysteine, whereas variable modification included oxidation at methionine. Search results from X!Tandem and Open Mass Spectrometry Search Algorithm were further analyzed through Trans-Proteomic Pipeline (TPP, version 4.6.0) (Deutsch et al, 2010) using PeptideProphet and iProphet, followed by SWATH assay library building procedures as detailed previously (Schubert et al, 2015; Guo et al, 2015). Altogether, we identified 160,442 peptides with <1% false discovery rate.

## Peptide quantification using OpenSWATH

SWATH files were analyzed using the prostate tissue proteome assay library described above and OpenSWATH software as described previously (Röst et al, 2014). Briefly, wiff files were converted into mzXML files using ProteoWizard msconvert v.3.0.3316, and then mzML files using OpenMS (Sturm et al, 2008) tool FileConverter. OpenSWATH was performed using the tool OpenSWATHWorkflow with input files including the mzXML file, the TraML library file, and TraML file for iRT peptides. The false discovery rate for peptide identification was below 0.1%. High-confidence peptide features from different samples were aligned using the algorithm TRansition of Identification Confidence (version r238), which is available from https://pypi.python.org/pypi/msproteomicstools or https://code.google.com/p/msproteomicstools/. The following parameters for the feature_alignment.py are as follows: max_rt_diff = 30, method = global_best_overall, nr_high_conf_exp = 2, target_fdr = 0.001, use_score_filter = 1.

## Protein quantification

The concentration of each protein was quantified through the simultaneous measurement of several peptides. To optimize the protein quantification, we developed a new computational method, which combines maximally consistent peptides for each protein and excludes inconsistent (i.e., uncorrelated) peptides (Picotti et al, 2013). For example, variation of PTMs would result in peptide-level variation that is uncorrelated across samples because mostly only one of the two peptides would be affected by the PTM (Picotti et al, 2013). Given a set of peptides unambiguously assigned to a single protein, consistent peptides were selected using the following procedure: all pairwise correlations between all peptides of a protein across the samples were calculated at first. Peptide pairs with a Pearson correlation coefficient ($r$) of at least 0.3 were determined, resulting in clusters of correlated peptides. This procedure yielded one or more peptide clusters per protein. We used the largest cluster of each protein and we quantified the protein's concentration as the average intensity across the peptides in that cluster. The minimum cluster size was set to two, and proteins without a cluster of at least two correlated peptides were removed from the subsequent analysis. This procedure resulted in robust relative

quantification of 3,700 proteins with high correlation between technical replicates ($r \geq 0.95$) and no missing values.

## Determining the biological variance between punches in a specific tissue (intra-tissue variance)

Measurements of protein abundance differences between individual punches are affected by a combination of biological and technical factors. Thus, to quantify the biological variation between punches, we need to subtract the technical variance from the total variance, i.e., the combined variance due to technical and biological factors. Estimating the biological variance of protein levels between punches therefore requires estimates of the technical variance and the total variance. Intuitively, one would estimate both variances using a standard approach such as ANOVA in a single statistical model. However, technical replicates are paired because they come from the same punch and, thus, they are not independent, whereas the total variance needs to be estimated across punches, i.e., involving partially independent measurements.

Therefore, we decided to separately estimate technical and total variances. Here, technical variance was estimated from the dispersion of measurements between paired technical replicates and total variance was estimated from the dispersion of measurements between independent punches from the same specimen. Compared with an approach estimating both technical and total variance in a single statistical model, our approach has the caveat that the two variance estimates can be inconsistent in the sense that the estimated total variance can be smaller than the estimated technical variance. Obviously, this happens only for those proteins where the technical noise is large compared with the biological variance, in which case it is anyways impossible to reliably estimate the true biological variance (no matter which statistical approach is taken). We, therefore, conservatively accept that in those cases, we cannot provide an estimate of the biological variance. However, we assume that in most of those cases, the biological variance will be small compared with the other proteins for which we could estimate a biological variance.

In detail, the variances were estimated in the following way.

First, the protein concentrations (computed from peptide intensities as described above) were log10-transformed. Next, protein concentrations were quantile normalized per sample. As the signal distributions between non-tumorous (benign) and tumorous tissue (malignant: acinar and ductal) differed significantly, the normalization was performed separately for each tissue type. For each protein, we computed the technical variation for each sample and averaged the inter-replicate variance across all 30 samples (Tukey, 1977). Because technical replicates are (obviously) paired, the technical variance was estimated as the dispersion of the two replicates from their sample mean averaged across all punches ($n = 30$). Thus, the technical variance $\mathrm{VAR_{TECH}}$ of protein $i$ was estimated as follows:

$$\mathrm{VAR_{TECH_i}} = \frac{1}{n} \sum_{j=1}^{n} \frac{(x_{i,ja} - x_{i,jb})^2}{2},$$

with $x_{i,ja}$ and $x_{i,jb}$ being the two technical replicates ($a$ and $b$) of the protein-level measurements from punch $j$. In this case, no batch correction was performed because batch correction would reduce the technical variance (technical replicates were always in different batches), which might lead to underestimation of the technical variance. The final estimate of technical variances was computed after removing outliers above and below the 1.5 times interquartile range of 30 samples based on Tukey's method (Tukey, 1977).

The total variances between punches (i.e., the combined variance from technical noise and biological variance) were initially computed for each batch separately. Thus, variation among punches from the same specimen (same patient $p$ and same tissue type $t$) was averaged. Finally, total variances $\mathrm{VAR_{TOT}}$ between punches were averaged across batches

$$\mathrm{VAR_{TOT_i}}(p,t) = \frac{1}{2}\left[\mathrm{VAR}(x_{i,ja}, j \in P(p,t)) + \mathrm{VAR}(x_{i,jb}, j \in P(p,t))\right],$$

where $P(p,t)$ denotes all punches $j$ from patient $p$ and tissue $t$ (i.e., either benign, acinar, or ductal). The indices $a$ and $b$ denote the two technical replicates, as above. Thus, total variances were estimated purely from deviations within batches and are (unlike technical variances) not affected by batch-to-batch variation. As a consequence, technical variances are biased toward larger values compared with total variances. This approach is conservative in the sense that it minimizes the number of proteins that are falsely classified as having variable concentrations within tissues. Thus, this approach will likely underestimate the true number of proteins with large biological intra-tissue variance. Given the total variance and technical variance, the biological variance $\mathrm{VAR_{BIO}}$ of protein $i$ was computed as follows:

$$\mathrm{VAR_{BIO_i}}(p,t) = \mathrm{VAR_{TOT_i}}(p,t) - \mathrm{VAR_{TECH_i}}.$$

This scheme generated seven independent estimates of total variance per protein: four for patients 1 and 2 (benign and malignant acinar tissues) and three for patient 3 (benign, acinar, and ductal). The intra-tissue variance shown in Fig 4 is the average biological variance of a given protein across all patients and tissue types. The tissue-specific variances used for Fig 5 are the average variances across the patients for the respective tissues (benign, acinar, and ductal). The biological variance in tumor was estimated as the average of all acinar and the ductal (patient 3) tumor regions.

## Grouping of proteins and pathways based on their variability

In cases where the estimated technical variance is greater than the estimated total variance, subtracting the technical from the total variance yields a negative "variance estimate" (Fig S4). Because these negative "variances" are the result of our imperfect variance estimates, the distribution of these values can be used to quantify the inherent uncertainty in our estimates of the biological variance. Thus, we can use the distribution of the absolute values (the "mirror distribution" into the positive range) as a background distribution for the null hypothesis that the true biological variance is indistinguishable from zero (or that the total observed variance is exclusively due to technical variance). Based on this approach, 797 proteins had $P$-values below 0.01 and were, thus, classified as biologically variable proteins (i.e., significantly variable within the same specimen). These 797 variable proteins were further subclassified as follows: if the ratio of biological variance in benign to

biological variance in tumor was above 2, they were classified as "variable in non-tumor" (339 proteins); if the ratio of biological variance in tumor to biological variance in normal was above 2, proteins were classified as "variable in tumor" (93 proteins); 365 proteins with similar variances in both tissue types (i.e., not different by more than a factor of two) were classified as "variable in non-tumor and tumor." Stable proteins were defined by choosing the 100 proteins with the lowest biological variance. The remaining proteins, which were not assigned to any of the above four groups, were classified as "medium heterogeneity" proteins.

Note that our computation of empirical $P$-values for determining variable proteins is not critical for the conclusions. If we had simply chosen the top 200 most variable proteins (as the basis for groups 1–3) and compared them with the 200 most stable proteins (Group 4), the conclusions would be virtually identical.

### Gene set enrichment analysis

GO enrichment of proteins was performed using topGO, which takes the topology of the ontology into account. The enrichment analysis was carried out by using Fisher's exact test with the background of measured proteins in this study. We excluded GO terms with less than 10 proteins and with more than 300 proteins from the analysis (the former are too small and the latter are too generic). Furthermore, we reported only GO terms that had at least four proteins enriched (overlapping).

Intra-tissue heterogeneity of entire biochemical pathways was determined according to the protein-level variance. Pathway variability was calculated by averaging the biological variances of all proteins annotated for a given ConsensusPathDB pathway. We required that each pathway contained at least five quantified proteins. ConsensusPathDB combines pathway annotations from different sources. Thus, in some cases, the same pathway is reported more than once. In such a case, the pathway variant with the largest number of quantified proteins was used.

### Determining the variance between tissues (inter-tissue variance) and between patients (inter-patient variance)

Batch effects were corrected by centering each protein's concentration per batch. In our experimental design, batches were balanced in the sense that each batch had the same number of benign and malignant samples (three of each) and each batch had the same number of samples from the same patient (two patients per batch and three samples from each patient).

Inter-tissue variances were estimated using concentrations centered per patient (subtracting patient mean). Inter-patient variances were estimated using concentrations centered per tissue type (subtracting tissue mean across patients). All of those computations were based on batch-corrected concentrations and after averaging technical replicates. Batch-corrected values were also used for Fig 2.

### Patient cohort and TMA

The Ethics Committee of the Kanton St. Gallen, Switzerland, approved all procedures involving human materials used in this TMA, and each patient signed an informed consent. For the study, patients with BPH and matching ADCA were included, whereas advanced prostate cancer, infectious or inflammatory diseases, or other malignancies fulfilled exclusion criteria as described previously (Cima et al, 2011). A TMA was constructed using formalin-fixed, paraffin-embedded tissue samples derived from 83 patients (BPH, $n$ = 35; ADCA, $n$ = 48).

### Immunohistochemical staining and evaluation

The following primary antibodies were used to stain 4-μm slides of the TMA using the Ventana Benchmark (Roche Ventana Medical Systems, Inc.) automated staining system: ACTR1B (1:400; abcam, 60 min pretreatment), Desmin/DES (1:20; Dako A/S, 16 min pretreatment), KLK3/PSA (1: 10000; Dako A/S) and GDF15 (1:50; Biorbyt, 30 min pretreatment), ACPP (1:2000; DAKO A/S), ABCF1 (1:50; Novus Biologicals, 90 min pretreatment), NUP93 (1:50; Novus Biologicals, 60 min pretreatment), CUTA (1:100; Lifespan Biosciences, 60 min pretreatment), CRAT (1:100; Atlas Antibodies, 30 min pretreatment), and FSTL1 (1:100; Atlas Antibodies, 16 min pretreatment). Detection was performed with ChromoMap kit (Ventana) for ABCF1, PCP4, and CUTA and with OptiView DAB kit (Ventana) for the others (Desmin, KLK3/PSA, NUP93, CRAT, FSTL1, and PAP) using the heat-induced epitope retrieval Cell Conditioning 1 solution. Slides were counterstained with hematoxylin (Ventana), dehydrated, and mounted. For GDF15, 4-μm slides were stained using the Leica Bond (Leica Biosystems) automated staining system. For detection, the Bond Polymer Refine Detection kit and heat-induced epitope retrieval HIER2 solution (Leica Biosystems) following hematoxylin counterstaining was used. Staining intensities for each antibody were evaluated in a semiquantitative, four-tier manner (negative = 0, weak = 1, moderate = 2, and strong = 3), along with the occupied area (in 1%, 3%, 5%, and above 10% steps), by one pathologist (NJ Rupp). An immunoreactivity score (staining intensity multiplied by percentage of spot) similar to the recommendations by Remmele & Stegner (1987) consisting of "staining intensity × area (%)" was calculated.

## Data deposition

The SWATH raw data and analyzed data, as well as the assay library, are deposited in PRoteomics IDEntifications (Vizcaíno et al, 2014). Project accession: PXD003497.

## Supplementary Information

## Acknowledgements

This work was supported by the SystemsX.ch project PhosphoNet PPM (to R Aebersold and PJ Wild), the Swiss National Science Foundation (grant no. 3100A0-688 107679 to R Aebersold), the Foundation for Research in Science and the Humanities at the University of Zurich (to PJ Wild), the European

Research Council (grant nos. ERC-2008-AdG 233226 and ERC-2014-AdG670821 to R Aebersold), the German Federal Ministry of Education and Research (grants: Sybacol & PhosphoNetPPM to L Li and A Beyer), European Union's Horizon 2020 research and innovation programme under grant agreement no. 668858, and the Swiss State Secretariat for Education, Research and Innovation under contract number 15.0324-2 (to R Aebersold and PJ Wild). We thank OL Kon for critical reading of the manuscript.

## Author Contributions

T Guo: data curation, software, formal analysis, validation, investigation, visualization, methodology, writing—original draft, project administration, and writing—review and editing.
L Li: data curation, software, formal analysis, validation, investigation, visualization, methodology, and writing—original draft, review, and editing.
Q Zhong: data curation, formal analysis, visualization, and writing—original draft.
NJ Rupp: data curation, formal analysis, and validation.
K Charmpi: data curation, formal analysis, and visualization.
CE Wong: formal analysis and validation.
U Wagner: formal analysis and validation.
JH Rueschoff: formal analysis and validation.
W Jochum: formal analysis and validation.
CD Fankhauser: formal analysis and validation.
K Saba: formal analysis and validation.
C Poyet: formal analysis and validation.
PJ Wild: conceptualization, resources, data curation, supervision, funding acquisition, investigation, methodology, writing—original draft, project administration, and writing—review and editing.
R Aebersold: conceptualization, resources, software, formal analysis, supervision, funding acquisition, writing—original draft, project administration, and writing—review and editing.
A Beyer: conceptualization, resources, software, formal analysis, supervision, funding acquisition, validation, investigation, visualization, and methodology.

## Conflict of Interest Statement

R Aebersold holds shares of Biognosys AG, which operates in the field covered by the article. The research group of R Aebersold is supported by SCIEX, which provides access to prototype instrumentation, and Pressure Biosciences, which provides access to advanced sample preparation instrumentation. The remaining authors declare that they have no conflict of interest.

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
