## [Reviewer comments · Life Science Alliance]

Multi-region proteome analysis quantifies spatial heterogeneity of prostate tissue biomarkers

Tiannan Guo, Li Li, Qing Zhong, Niels J. Rupp, Konstantina Charmpi, Christine E. Wong, Ulrich Wagner, Jan H. Rueschoff, Wolfram Jochum, Christian Daniel Fankhauser, Karim Saba, Cedric Poyet, Peter J. Wild, Ruedi Aebersold, Andreas Beyer

DOI: 10.26508/lsa.201800042

Review timeline:	Submission date:	27 February 2018
	1 st Editorial Decision:	27 February 2018
	1st Revision Received:	3 April 2018
	2 nd Editorial Decision:	15 April 2018
	2 nd Revision Received:	11 May 2018
	Accepted:	16 May 2018

Report:

(Note: Letters and reports are not edited. The original formatting of letters and referee reports may not be reflected in this compilation.)

1st Editorial Decision 27 February 2018

Thank you for transferring your manuscript entitled "Multi-region proteome analysis quantifies spatial heterogeneity of prostate tissue biomarkers" to Life Science Alliance. The manuscript was assessed by expert reviewers at another journal and the reports have been transferred to us.

I have discussed your work with an academic editor in light of the reports you have obtained, and we would like to publish your work as a resource article in Life Science Alliance, pending satisfactory minor revision.

A revision of your work seems straightforward in our view. Importantly, please present your data in a way that will enable others to easily access and re-use the data (see also report of referee #2). Note that we don't insist on comparison of your proteomics data with RNA-seq data, and that we will ask referee #1 and #3 to re-evaluate the revised version of your work.

Please let us know in case you would like to discuss individual revision points further.

Thank you for this interesting contribution to Life Science Alliance. We are looking forward to receiving your revised manuscript.

1st Revision – authors' response 3 April 2018

Referee #1 (Comments on Novelty/Model System for Author):

The authors applied high pressure cycling technology to extract protein from tissue samples, followed by protein reduction, alkylation, Lys-C/Trypsin digestion, and LC-MS/MS analysis to identify/quantify proteins in multi-region of 60 biopsy-level tissue samples from three patients with prostate cancer. The goal of this study is to systematically investigate the proteome spatial intra-

tissue heterogeneity (ITH) based on the evident of large genomic heterogeneity observed. Over 8,000 proteins were quantified, and 3,700 proteins were found with ITH. Based on the ITH discovery study, the authors further validate their results by immunohistochemistry (IHC) analysis of tissue microarrays (TMA). The benign and malignant (ADCA) prostate tissues from 83 additional patients were selected for the validation study. The protein expression from ten representative proteins in the various ITH groups identified from the discovery study were measured, including ACTR1B, DES, PSA, GDF15, ACPP, ABCF1, NUP93, CUTA, CRAT, and FSTL1. The results from validation study further confirmed the findings of discovery study, including the PSA was more variable in benign prostatic hyperplasia, while GDF15 were varied in different tumor regions. This study demonstrates that the expression of protein markers such as PSA and ACPP in prostate cancer tissue has spatial intra-tissue heterogeneity, providing guidance for clinical application of by adding extra criterion for the assessment of the protein biomarker candidates.

Following suggestions for authors:

1. The protein markers such as PSA, DES, and PAP in both the discovery study and validation study have different PTMs including phosphorylation and glycosylation. It will be very important and critical to quantitatively analysis of those protein markers with ITH at certain protein PTM form.

Reply: We agree that PTMs are playing critical roles in biological processes. There are two aspects to this problem: first of all, PTMs may change the observed peptide intensities, which could obscure true protein level variation (i.e. peptide intensities vary between locations due to different PTMs rather than protein level differences). Second, PTMs as such are important modifications affecting protein activities. Addressing the functional role of spatial PTM heterogeneity (i.e. addressing the second point) is beyond the scope of this manuscript. The focus of this study lies on protein levels and studying spatial variability of PTMs requires a different experimental and computational approach. However, we do need to account for potential peptide level changes due to PTMs rather than protein level changes. We have done this in our study by requiring that each protein is quantified by two correlated peptides. A PTM would most likely affect only one of the two peptides and thus lead to diverging intensity patterns of the two peptides across samples. Hence, our approach of robust protein quantification at least partly addresses the PTM issue.

2. Some of protein biomarkers in the validation study are secreted proteins such as PSA CUTA, GDF15, and ACPP. Their expression measured by TMA assay may not truly reflect the status of ITH since they may be truncated through cleavage.

Reply: Thank you for this comment. We have added discussion of this limitation of TMA in the manuscript. The new texts are copied below:

“Nevertheless, it is worth noting that TMA may not truly reflect the exact protein expression because secreted proteins will be lost, which partly explains the small discrepancy between TMA and MS data.”

3. It will be important to provide mRNA expression level for the ten protein markers used in the validation study so that we can understand if there is correlation between protein expression levels and mRNA levels when probe the ITH.

Reply: While we agree with the reviewer that comparing the protein heterogeneity and the mRNA heterogeneity is an interesting research question, we respectfully maintain that it is out of the scope of the current study because our study is focused to analyze the heterogeneity of proteins which are the relevant molecular entities for clinical biomarkers. Our unpublished data showed that mRNA suffers from degradation in the clinical specimens we have analyzed. A more sophisticated model is required to compare mRNA expression and protein expression in clinical specimens, which we regard as out of the scope of the current manuscript.

Referee #2 (Remarks for Author):

The primary focus of this carefully conducted study by Guo et al. is to evaluate the intertumoral heterogeneity (ITH) of proteins (rather than genes or transcripts) in context of prostate cancer and benign prostatic hypertrophy occurring within the same tissue material. The authors use SWATH, a tradename for data-independent analysis (DIA) mass spectrometry data, to collect and analyze data

for ca. 3700 proteins from multiple cores obtained from tissue of three patient samples. The results are interesting, but largely confirmatory at the protein-level of prior knowledge about tissue and tumor heterogeneity obtained by genomics. As such, the paper is largely a technical evaluation of a specific data collection approach in mass spectrometry-based proteomics. That normal and/or benign tissue shows higher inter-core variability than tumor tissue is not surprising given the complexity and variability of the extracellular matrix. Some new and interesting results derive from GO and pathway-level annotation of the proteins observed that are either stable or variable in expression levels in tumor and non-tumor and variable between tumor and BPH (Figure 4). Perhaps the most interesting observation is that proteins involved in DNA repair are found to be heterogenous in tumor tissue, as already established by genomic methods.

Issues to be addressed:

- Because it is very unlikely that readers will download and attempt to use the authors software that is required to be used to interrogate the underlying data, the authors should improve content of their supplemental tables to enable one to easily identify what peptides for each protein are being claimed to be confidently identified. Specifically:
- Supplementary table 3 should be reorganized to present peptides organized by the protein from which they originate. Both protein ids and gene names and accession numbers should be provided
- Supplementary table 4 is currently uninterpretable as codes rather than protein ids and gene names and accession numbers are being used. Both should be supplied for all entries, and the two

Reply: We have added gene names and IDs in the supplementary Tables 3 and 4. In addition, to avoid ambiguity of protein groups which contain multiple genes, we keep only the protein groups with a single gene symbol based on the SwissProt database.

- Authors should comment in text as to whether any of the following key mediators of DNA damage response in prostate cancer were detected in their analyses: BRCA1, BRCA2, PALB2, ATM, ATR, RAD51, MRE11, CHEK2, and XRCC2/3. If not, why not?

Reply: We have added the following sentences in the Discussion.

“We quantified the degree of ITH of several key proteins involved in DNA damage response, including ATR, MRE11, RAD21, RAD23A, RAD23B, RAD50, RAD9A, CHEK1, XRCC5 and XRCC6. The data showed that ATR, a DNA damage sensor, is variable only in tumors. We identified more proteins in the 6873-protein list which contains at least one proteotypic peptide, including BRCA2, ATM, RAD51C, RAD51AP2, XRCC1, XRCC2, XRCC4. However, these proteins were not included in the ITH analysis due to either single-proteotypic-peptide identification or discordant quantity of multiple proteotypic peptides in a protein that failed to pass our stringent inclusion criteria.”

- amount of protein analyzed/core sample should be clearly stated - not just dimensions.

Reply: We have provided the tissue weight in the Methods. It is ~2 mg wet weight. We also added in the Methods: “About 100 µg proteins and 50 µg peptides were extracted per mg tissue.”

- The variability of myosins, a very large family of structurally-related proteins, in prostate is well known (e.g., see Cell Rep. 2015 Dec 15; 13(10): 2118-2125) and Myo1b and Myo10 have been shown to be expressed at higher levels in metastatic tumors. The authors should comment on whether myosins, specifically certain family members, were detected and could distinguish tumor from non-tumor in their samples. If myosin family members could not be readily distinguished, this should be commented on in text.

Reply: We thank the reviewer for the comment. In the referred study, the authors compared myosin expression in different prostate cancer cell lines (LNCaP, DU145 and PC-3) and matched prostate cell lines (1535NP and 1535CT). They observed Myo1b, Myo9b, Myo10, and Myo18a were expressed at higher level in cell lines with higher metastatic potential (PC-3). In our data, we have identified from prostate tissues multiple myosins including MYO1A, MYO1B, MYO1C, MYO1D, MYO1F, MYO1G, MYO5A, MYO5B, MYO6, MYO7B, MYO9A, MYO9B, MYO18A, MYO18B, and MYO19. However, our data can not be compared with this referred study, because our study is not aimed to detect proteins that can distinguish tumorous from non-tumorous samples. The goal of

our study is to compute the spatial variability of proteins, not their fold-change between tumors and non-tumors. For instance, a protein can be highly variable in different regions of a prostate tissue of specific histology, but no fold-change between tumorous and non-tumorous tissues. On the other hand, a protein that is a cancer biomarker can be of low degree of intra-tumor heterogeneity, or vice versa. We respectfully maintain that these are two orthogonal dimensions of the protein expression that should not be mixed up, therefore, our data can not support the findings from this myosin reference paper.

- As far as this reviewer can tell, the claim that 8,248 proteins were confidently identified is based on including single peptide identifications. The 3700 number that used 2-peptides/protein is probably closer to reality. The correct number based on 2-peptides/protein should be stated.

Reply: We have revised the abstract. The new text reads:

“We quantified 8,248 proteins and analyzed the ITH of 3,700 proteins confidently quantified by at least two proteotypic peptides.”

- The comparison in Discussion of their study to recently published studies by NCI-funded proteomics initiatives is a bit misleading. The latter studies used the 2-peptide/protein criteria and identified 8000 - 9000 proteins/patient samples, albeit with considerable fractionation and instrument time.

Reply: We have removed the comparison of our data and the NCI-funded proteomics initiatives, because the protein number is not directly comparable. These studies used different protein sequence databases, different protein inference algorithm (we only considered proteotypic proteins while these NCI-funded projects used protein groups). The new text reads:

“Despite the rigorous filtering, we could quantify a three times higher number of proteins than a recent proteomic analysis of primary prostate tissue samples (Iglesias-Gato, Wikstrom et al., 2016).”

- The authors clearly state that their study was not aimed at identifying new proteins that could be used as biomarkers, but rather to characterize spatial heterogeneity. However, the only way to address the ITH issue is to take multiple needle core biopsies from each patient's tumor. Is this clinically acceptable? Authors should comment.

Reply: We have, indeed, taken multiple core needle biopsies from each tumor as shown in Figure 1. However, our intention is not to quantify ITH routinely in prostate cancer patients. Instead, we aimed to understand the biology of protein heterogeneity in prostate cancer and we propose to consider ITH in future biomarker studies for the identification of robust markers. Having that said, taking multiple core needle biopsies from a single tumor is becoming routine.

Prostate cancer patients currently receive a minimum of 12 core needle biopsies (six from both sides of the prostate). In certified prostate cancer centers, e.g. at the University Hospital Zurich, Switzerland, many patients with limited disease are treated locally by high-frequency ultrasound (HIFU). In order to increase the diagnostic accuracy of multiparametric MRI and fusion-guided targeted biopsy, so called transperineal template saturation prostate biopsies are performed for the detection and characterization of prostate cancer (Mortezavi et al., J. Urol. 2018). Up to 48 core needle biopsies or even more are taken in such a setting to systematically generate a map of the infiltrating adenocarcinoma before decision making.

-While the present study identified many more proteins, the key proteins associated with intra and inter tumoral heterogeneity can and were all measured by IHC. Authors should also comment that if multiple biopsies were taken, would IHC then provide sufficient information as to the extent of ITC and better inform clinical decisions.

Reply: We have added the following comment in the discussion:

“Nevertheless, it is worth noting that TMA may not truly reflect the exact protein expression because secreted proteins will likely be lost during IHC procedures, which partly explains the small discrepancy between TMA and MS data. In addition, not every protein can be measured by TMA and IHC depending on the availability of proper antibody.”

- A recent paper by Buczak et al. conducted a quantitative proteomics study of inter- and intra-tumor

protein heterogeneity in context of hepatocellular carcinoma and demonstrated the value of proteomics for studying spatial heterogeneity (Buczak et al. Mol Cell Proteomics mcp.RA117.000189. First Published on January 23, 2018, doi:10.1074/mcp.RA117.000189). This paper should be cited. In addition, the authors of this work compared what could be learned by RNA-seq vs. proteomics. Authors should comment on the relative merits of the transcript data vs proteomics data in context of their study.

Reply: We have included this reference and comments in our Introduction. The texts are copied below:

“Label-free shotgun proteomics has been used to compare the proteomes of three regions of colon tissues (Wisniewski, Ostasiewicz et al., 2012). During the review of this study, Buczak, et al reported quantitative proteomic comparison of five pairs of tumorous and non-tumorous micro-dissected FFPE tissues from patients with hepatocellular carcinoma (HCC) using 10-plex TMT, and identified protein abundance changes between tumorous and peri-tumorous tissues including NADH hydrogenease complex I which is also observed as changed in 11 murine HCC tumors compared to normal murine livers using label-free quantification (Buczak, Ori et al., 2018). In another experiment of three concentric sector regions, a tumor capsule region, a peritumoral tissue region and the bulk tumor, the authors quantified 2698 Uniprot proteins (excluding protein groups) using 6-plex TMT and 2166 proteins using DIA. This study found the majority of the quantified proteins were expressed at comparable levels across the whole specimen, and detected abundance changes of multiple proteins across regions including collagens, Fibrillin and Decorin. The authors also identified consistency between proteome and transcriptome data in terms of gene expression changes, implying that spatial heterogeneity is largely driven by protein synthesis variation. Despite this progress, it remains important to separate technical variability from true spatial ITH and to investigate the relationship between inter-tumor heterogeneity and intra-tumor heterogeneity. Answering these questions requires a rigorously designed study, a highly reproducible proteomics technology, the ability to analyze multiple regions of a bulk tumor, and statistical models to deconvolute various types of protein variation.”

- Statements like "very robust concentration estimates" are not substantiated and should be deleted unless adequate proof of both reproducibility and accuracy of protein concentration is provided.

Reply: we have changed "very robust concentration estimates" to “robust relative quantification of 3,700 proteins”.

Referee #3 (Remarks for Author):

This study aims to explore the magnitude of proteome intra-tissue heterogeneity (ITH) in tumor tissues. Using SWATH-based quantitative proteomics, the authors analyzed several punches of benign and malignant (acinar) regions within the same prostate cancer section, from various patients. To determine biological variance in protein expression between closely spaced punches, the authors subtracted technical variance from total variance for each measured protein. Among 3700 proteins that were quantified by at least 2 proteotypic peptides, several hundred showed heterogeneous spatial expression in benign or malignant tissue, or both. They perform GO and pathway analysis on each category, and identify immune-associated processes to be heterogeneous in both tissue types, while muscle and chromatin-related proteins varied more in benign and malignant tissue, respectively. They confirm heterogeneous expression of some of the identified proteins by immunohistochemistry.

While most studies in cancer proteomics focus on the identification of proteins that are differentially expressed between normal and malignant tissue, or between tumor types/grades (i.e. proteins that may serve as disease biomarkers), this manuscript instead aims to quantify heterogeneity in protein expression within regions that had been classified as benign or malignant by classical tissue staining. The authors take great care in their statistical analysis to differentiate technical from biological variance, and this is the stronger part of the manuscript. The weaker part resides in the explanation or interpretation of the found heterogeneity. In particular, the authors perform GO and pathway analyses for proteins with different degrees of expression variance (Fig 4), however the authors tend to read too much into these data, e.g. 'variability in DNA repair pathways ... may contribute to the genetic heterogeneity' (line 38-39), and detection of proteins involved in nucleosome and chromatin

assembly are interpreted as differences in proliferation rate (line 229-230). This is all highly preliminary and up for many other explanations, so authors should be cautious to not over-interpret. Furthermore, with this gained knowledge, what does this mean for clinical practice - even just conceptually? For instance, does protein expression variance in malignant tissue matter a lot if its overall expression level is much higher (or lower) than in healthy tissue?

Reply: This study aims to address a critical, unaddressed issue in clinical diagnosis based on protein measurement of biopsy samples. We have designed an experiment specifically to deconvolve the various layers of variability of protein measurement, developed rigorous statistical analyses of these variabilities, stratified 3700 proteins based on their spatial heterogeneity. We then performed targeted analysis of a few widely used or promising biomarkers including PSA and GDF15 using tissue microarray and an independent cohort. We have made this point clearer by modifying the last sentence of the Abstract, as copied below:

“This study suggests that the spatial variability of proteins should be taken into account when they are utilized as biomarkers. We stratified the prostate proteome based on their spatial heterogeneity. ITH is protein-dependent.”

The second major concern is that it remains unclear how proteome variance, as a characteristic that is distinct from mere presence or expression level of a protein. This becomes somewhat blurred by the mixed use of 'variance' and 'heterogeneity'. Although these terms are used for a high-level interpretation of the data, they go past the more down-to-earth conclusion that protein expression is different between the tissue punches even if they originate from the same pathological zone. By inference, this means that taking just 3 punches per zone is at the lower limit to detect these differences, and that more will be needed to increase the resolution and potentially biological insight.

Overall, given the focus on developing the methodological framework with only preliminary biological implications, maybe the take-home message of this study is that protein expression within pathologically-classified areas is heterogeneous, and that this can be measured by SWATH-MS.

Reply: As addressed in the previous Reply, the take-home-message of this study is that “the spatial variability of proteins should be considered when they are utilized as biomarkers”. In addition, “We stratified the prostate proteome based on their spatial heterogeneity.” We observed that proteins have different degree of ITH. ITH should be considered for each individual protein. Specifically, we identified and validated ITH pattern of PSA and GDF15.

Other remarks:

1) The authors mention a number of alternative techniques to analyse proteome heterogeneity, but miss two important ones: the first is tissue micro-dissection combined with MS, which should be mentioned especially because of its emergence due to increased sensitivity of proteomic workflows (e.g. PMID 29133510, 29363612, 28358042). Second, and even more importantly, MALDI imaging is potentially a much more powerful approach than SWATH (or other LC-MS-based methods) when it comes to assessing tissue heterogeneity: although it may not detect as many proteins or peptides as LC-MS, its resolution and speed are unsurpassed in detecting fine-grained differences within tissue sections (e.g. PMID 27168173 and 25201776 to mention a few). This should be mentioned and contrasted to SWATH-MS.

Reply: We have added discussion of the laser capture microdissection and MALDI imaging references in the Discussion, as copied below:

“Our workflow is also compatible with laser capture microdissected samples which can also be analyzed by shotgun proteomics (Buczak et al., 2018, Garcia-Berrocso, Llombart et al., 2018, Grosserueschkamp, Bracht et al., 2017).”

“Advanced MALDI imaging emerges as a useful tool to dissect ITH of proteins, peptides and small molecules with high spatial resolution (Balluff, Frese et al., 2015, Widlak, Mrukwa et al., 2016), however, the proteome depth and precision await further improvement.”

2) With the knowledge of the 3700 proteins meeting the analytical thresholds to be considered for

detailed analysis, would it be possible to revisit the remainder of the 8200 identified proteins to seek confirmation/refinement of the biological conclusions?

Reply: We thank the reviewer for the advice, however, we maintain that high quality of quantitative proteomics data is essential for the deconvolution of protein variability. To meet the high quantitative data quality, we designed and performed the experiment in a way that different layers of technical variability and biological variability can be rigorously estimated and isolated. Future algorithms may be able to dig out more information from the remaining proteins, however, we think this is beyond the scope of the current study. With the current bioinformatics ability, we could only reliably analyze the ITH of 3700 proteins.

3) Line 194-199: the authors use numbers in the text while using colors in Fig 4 to indicate protein expression categories. Mentioning the numbers in the figure will be very helpful to follow the discussion.

Reply: Thank you for the advice. We have modified **Figure 4A** accordingly.

2nd Editorial Decision

15 April 2018

Thank you for submitting your revised manuscript entitled "Multi-region proteome analysis quantifies spatial heterogeneity of prostate tissue biomarkers".

Two of the reviewers who evaluated your study at another journal before have now commented on the revised version. As you will see, both reviewers are satisfied with the revision performed and support publication of your work in Life Science Alliance. We would thus be happy to publish your paper pending final revisions necessary to meet our formatting guidelines. Congratulations on this very nice work!

 REFEREE REPORTS

Reviewer #1 (Comments to the Authors (Required)):

The authors have done their best to modify the manuscript in response to reviewers' comments. Though not all issues have been thoroughly addressed, the paper as it is does represent a significant contribution to the literature and approval is recommended

Reviewer #3 (Comments to the Authors (Required)):

The authors have satisfactorily addressed my concerns and I recommend publication in Life Science Alliance.

Accepted

16 May 2018

Thank you for submitting your Resource entitled "Multi-region proteome analysis quantifies spatial heterogeneity of prostate tissue biomarkers". It is a pleasure to let you know that your manuscript is now accepted for publication in Life Science Alliance. Congratulations on this interesting work.